# Assessment of PM2.5 Concentration at University Transit Bus Stops Using Low-Cost Aerosol Monitors by Student Commuters

**DOI:** 10.3390/s24144520

**Published:** 2024-07-12

**Authors:** Will Murray, Qiang Wu, Jo Anne G. Balanay, Sinan Sousan

**Affiliations:** 1Department of Public Health, Brody School of Medicine, East Carolina University, Greenville, NC 27858, USA; murrayjo20@students.ecu.edu (W.M.); wuq@ecu.edu (Q.W.); 2Environmental Health Sciences Program, Department of Health Education and Promotion, East Carolina University, Greenville, NC 27858, USA; balanayj@ecu.edu; 3North Carolina Agromedicine Institute, Greenville, NC 27834, USA; 4Center for Human Health and the Environment, NC State University, Raleigh, NC 27695, USA

**Keywords:** PM_2.5_, particulate matter, air quality, aerosol personal exposure, low-cost aerosol monitor

## Abstract

Particulate matter of 2.5 µm and smaller (PM_2.5_) is known to cause many respiratory health problems, such as asthma and heart disease. A primary source of PM_2.5_ is emissions from cars, trucks, and buses. Emissions from university transit bus systems could create zones of high PM_2.5_ concentration at their bus stops. This work recruited seven university students who regularly utilized the transit system to use a low-cost personal aerosol monitor (AirBeam) each time they arrived at a campus bus stop. Each participant measured PM_2.5_ concentrations every time they were at a transit-served bus stop over four weeks. PM_2.5_ concentration data from the AirBeam were compared with an ADR-1500 high-cost monitor and EPA PM_2.5_ reference measurements. This methodology allowed for identifying higher-than-average concentration zones at the transit bus stops compared to average measurements for the county. By increasing access to microenvironmental data, this project can contribute to public health efforts of personal protection and prevention by allowing individuals to measure and understand their exposure to PM_2.5_ at the bus stop. This work can also aid commuters, especially those with pre-existing conditions who use public transportation, in making more informed health decisions and better protecting themselves against new or worsening respiratory conditions.

## 1. Introduction

Fine particulate matter (PM_2.5_) refers to particles 2.5 µm or less in aerodynamic diameter that can could originate from vehicle exhaust, the burning of fuel such as wood, oil, or coal, and smoke from fires [1]. PM_2.5_ in high concentrations is known to cause respiratory issues, including irritation of the airways, sneezing, coughing, and difficulty breathing [2,3]. Due to their small particle size, PM_2.5_ can reach certain depths of the respiratory system that larger particles cannot. As a result, PM_2.5_ exposure can have a much more severe effect on individuals with pre-existing respiratory conditions such as asthma, chronic obstructive pulmonary disease (COPD), and heart disease, potentially causing a worsening of the condition, hospitalization, and even death [2,4,5]. Given the damaging health effects of exposure to high concentrations of PM_2.5_, it is especially important for those with pre-existing respiratory conditions to be aware of their risk of exposure in daily activities.

The United States Environmental Protection Agency (EPA) has in place a daily PM_2.5_ standard of 35 µg/m^3^. To monitor ambient PM_2.5_ concentrations, the EPA operates numerous air quality monitoring sites throughout the United States, 48 of which are in North Carolina [6]. Ambient air monitoring is performed at these sites through the federal reference method (FRM) or federal equivalent method (FEM), which are designed to provide the most scientifically accurate measurements possible [7]. These monitors perform gravimetric filter analysis, which provides a daily average of particulate matter concentration. The EPA’s federal equivalent method (FEM) monitoring stations can provide high-frequency temporal measurements during the day [7]. However, the number of FRM and FEM monitoring sites may be inadequate given the size of the geographical areas they represent, and the instruments are usually deployed on a county level with less dense populations [8]. Thus, the air quality information used on the county or city level may not allow residents to assess their daily exposure risk and may not truly represent the geographical areas where they live [9]. In relation to ambient PM_2.5_ concentrations and respiratory health, exposure misclassification can cause EPA-reported particulate matter concentrations, a factor often used by individuals with respiratory conditions in personal health decision-making, to be unreliable or inaccurate [10]. Despite the EPA’s efforts, one air quality monitoring station may not be sufficient to provide comprehensive, location-specific air quality and PM_2.5_ concentration information for the county [9].

Alternative high-cost aerosol instruments like the ADR-1500 (Thermo Scientific, Franklin, MA, USA) measure PM_2.5_ concentration using highly sensitive light-scattering photometer technology [11]. While monitors like the ADR-1500 are highly accurate and may serve as the standard to which others are compared, they can cost thousands of dollars or more. Additionally, the complexity and sensitivity of the technology used to measure particulate matter concentration necessitates use only by trained professionals, so these monitors cannot be deployed in large quantities [12]. To increase the accessibility of real-time air quality data and to better account for exposure misclassification, low-cost aerosol monitors have become more frequently utilized by individuals and researchers alike [13]. These monitors are significantly less expensive, more lightweight, and easier to operate than instruments like the ADR-1500. Low-cost aerosol monitors can provide individuals with PM_2.5_ measurements that are more relevant to their specific environment and personal health [14]. However, due to differences in technology between reference instruments and low-cost monitors, measurements taken by low-cost monitors are essentially estimates of ambient PM_2.5_ concentrations. To ensure that these measurements reflect actual particulate matter concentration, low-cost sensors must be calibrated alongside a reference instrument [15]. Overall, low-cost aerosol monitors are an accessible, affordable, and portable method for individuals with respiratory conditions to assess PM_2.5_ exposure in real time. These sensors have been used and extensively evaluated in many studies related to environmental, occupational, indoor, personal, and laboratory conditions [16,17,18,19,20,21].

The increase in popularity, utilization, and development of low-cost aerosol sensors has facilitated extensive research to examine sensor accuracy through a calibration process. The AirBeam (HabitatMap, Brooklyn, NY, USA) is a low-cost optical particle counter that measures multiple particle sizes, including PM_2.5_ [22]. The company has released three versions of the AirBeam that have each been evaluated in laboratory, environmental, and occupational settings [23,24]. The company has shown that the AirBeam3 has more accurate measurements compared to the AirBeam2 when compared to a FEM [25]. In addition, the South Coast Air Quality Management District evaluated versions 1, 2, and 3 and proved that the AirBeam3 was more accurate than any previous generation of AirBeam compared to FEM monitors [26,27,28]. 

While sustained exposure to high concentrations of fine particulate matter may be common and accounted for in certain environments or workplaces, college students are not typically regarded as a population that would receive this type of exposure. Many college students rely on public or university-provided transit systems to commute to and from class, campus buildings, and their residences. Previous studies have examined PM_2.5_ exposure in public transport systems and students [29,30,31]. Hess et al. [32] evaluated the exposure of commuters at seven bus stations in Buffalo, New York; the researchers determined that commuters were exposed to a higher mean PM_2.5_ concentration inside the bus shelters (17.24 µg/m^3^) than outside (14.72 µg/m^3^) and that PM_2.5_ concentration was higher in the mornings (18.84 µg/m^3^) than in the afternoons (13.08 µg/m^3^). Commuters at the bus stations in the study were exposed to a mean PM_2.5_ concentration of 15.98 µg/m^3^. As for bus stops specifically, Qiu and Cao [33] determined that commuters in Xi’an, China, were exposed to a mean PM_2.5_ concentration of 72.4 µg/m^3^ while waiting at roadside bus stops. However, no research has yet been published on the effects of university transit systems on personal PM_2.5_ concentrations at bus stops and the associated exposure risk of college students. 

The objective of this research study was to measure the personal exposure of students to PM_2.5_ at different university transit bus stops. Measurements collected from the bus stops were analyzed alongside data from an EPA FEM site and a high-cost aerosol instrument located in the same city to determine whether students who use the transit bus system were exposed to PM_2.5_ concentrations higher than those reported by the EPA and the high-cost instrument. Additionally, this study examined any potential trends in PM_2.5_ concentration across various geographical locations and bus stops.

## 2. Materials and Methods

### 2.1. Reference and High-Cost Instruments 

This study utilized data collected from the FEM (BAM-1022, Met One, Grants Pass, OR, USA) real-time monitor managed by the EPA on the county level and located in the same city. The EPA site is located in a light-traffic area with small parking lots within 483 km of the site. Additionally, this work used a deployed ADR-1500 (Thermo Scientific, Franklin, MA, USA), a high-cost (>$10,000) real-time aerosol monitor. The ADR-1500 was located at the intersection of two high-traffic four-lane main streets in Greenville, NC. The ADR-1500 was set to measure PM_2.5_ concentrations using a cyclone at a constant flow rate of 1.52 LPM. PM_2.5_ concentrations were recorded daily with the ADR-1500, and the EPA website provided daily concentration data on a regular basis for public use. The ADR-1500 can perform 37 mm filter measurements that can be used for gravimetric analysis and correction of the ADR-1500 measurements. In addition, the ADR-1500 incorporates a heater to dry the air stream, eliminating biases due to the effects of humidity. However, filter measurements were not provided for the ADR-1500 at the time of this study, so the FEM measurements were also used to compare between the three technologies. 

### 2.2. Low-Cost Aerosol Monitors

The low-cost personal PM_2.5_ monitors used in this study were the AirBeam2 and AirBeam3 (Habitatmap, Brooklyn, NY, USA), which are shown attached to a student’s backpack in Figure 1. The AirBeam2 and AirBeam3 monitors were chosen for this study due to their relatively low cost (~$250), ease of use, and calibration for environmental exposure. Primarily, the AirBeam monitors have been calibrated by the manufacturer and extensively evaluated in outdoor environmental settings and have shown a high correlation (r^2^ ≥ 0.9) compared to reference instruments [26,28]. Therefore, calibration was not performed for the sensors in this study. Moreover, performing such calibration that entails deploying a high-cost instrument with low-cost sensors for personal exposure was impractical for this study due to the cost of the instruments and the inability to train students to operate these high-cost monitors, defeating the purpose of performing citizen science research. The purpose of this study is to use these manufacturer-calibrated and well-evaluated low-cost sensors to better understand student exposure at the bus stop and show the differences by comparing these values with reference and high-cost instruments deployed in the same city and county while considering their limitations based on previous work. This is important because EPA sites represent county-level data, and this work shows that averaging data for the county presents biases. 

Three AirBeam2 monitors were available from previous work, and four AirBeam3 monitors were purchased for this study. The AirBeam2 and AirBeam3 use PMS7003 sensors (Plantower Technology, Nanchang City, Jiangxi, China). The AirBeam aerosol monitors can be used to measure the ambient concentration of PM1 (particles 1 µm or less in size) and PM_2.5_. The monitors also measure temperature and relative humidity and provide the GPS location. The AirBeam uses light scattering technology to estimate the number of particles in the air, recording a PM_2.5_ concentration measurement once per second. The device itself weighs only 6 ounces and can easily be attached to a belt loop or bag for mobile monitoring. The AirBeam is accompanied by the AirCasting mobile software (Version 1.15.0). The AirBeam2 is compatible only with Android devices, whereas the AirBeam3 is compatible with both Android and iOS devices; the AirCasting mobile application can be downloaded from each device’s respective app store. Both monitors connect to the user’s mobile phone via Bluetooth. Air monitoring sessions are recorded and stored in the AirCasting mobile application.

### 2.3. Study Location, Participants, Field Deployment, and Data Collection

This study was conducted at East Carolina University (ECU) in Greenville, North Carolina. This university’s transit bus system provides upward of two and a half million rides per year, allowing students to travel around campus and the city of Greenville (ECU Parking & Transportation, 2023). The bus service is free for students and only requires the student’s university-issued 1Card to ride. ECU’s transit system comprises 28 GILLIG buses; 24 of these buses are diesel-powered, 2 are diesel/electric hybrids, and 2 are powered by compressed natural gas (CNG). Though the university has recently introduced these hybrid and CNG-powered buses to reduce emissions, most of its bus fleet is still powered by standard diesel fuel. A representative from ECU Transit stated that the university was actively working to replace more of their diesel-powered buses with hybrid and CNG-powered buses, although the COVID-19 pandemic had significantly impeded the transition. Today’s diesel-powered engines run cleaner than those of the past, but diesel fuel still emits significantly higher levels of PM_2.5_ compared to gasoline-powered engines [34]. ECU has thirty-five bus stops located on and around campus, most of which are located alongside busy roadways or large parking lots. The point could be made that students who do not use the ECU Transit bus system would not ordinarily spend any length of time stationary in a parking lot or on the side of a large highway. These factors, combined with a majority diesel-powered bus fleet and its associated emissions, could increase ECU students’ risk of exposure to high concentrations of PM_2.5_, particularly affecting those students with pre-existing respiratory conditions such as asthma.

A total of 7 ECU student participants were recruited for this study. Three AirBeam2 monitors and four AirBeam3 monitors were distributed amongst the participants. The first three recruited participants received an AirBeam2 monitor, while the remaining four participants were provided with AirBeam3 monitors. Each participant was trained on the proper use of the monitors and the process of recording PM_2.5_ concentration measurements with the AirCasting mobile application installed on their mobile phones. Each participant powered on their respective AirBeam monitor when they arrived at an ECU Transit bus stop; the monitor was allowed to run and record PM_2.5_ concentrations for the entire duration of the student’s waiting period at the bus stops, allowing the student’s personal exposure to be measured. Participants ended the recording session when they boarded the bus, and the monitor was powered off and stored securely in the participant’s bag. 

Each participant repeated the above procedure for four weeks. Due to variations in the dates on which each of the first three participants began and concluded their participation in this study, the date ranges of their four-week study periods also varied. The four subsequent participants began their participation in this study on the same day, thus completing their four-week study periods on the same day as well. In total, PM_2.5_ data were collected from 15 February 2023 to 14 April 2023. The date range when participants collected the data is shown in Table 1.

Because the primary objective of this study was to measure ECU students’ personal exposure to PM_2.5_, the research team analyzed each participant’s daily transit schedule prior to the start of this study and created a list of all bus stops used. At the conclusion of the study period, the seven participants had used a total of eight different ECU Transit bus stops. The location of the bus stops, ADR-1500, and EPA monitor are shown in Figure 2. To streamline data analysis, each bus stop was assigned an identification number. These bus stops included the Main Campus Student Center Hub (BS1), Christenbury Gym Hub (BS2), College Hill stop (BS3), West End stop (BS4), Carol Belk Building stop (BS5), Ficklen and Charles stop (BS6), Ficklen Drive stop (BS7), and Health Sciences stop (BS8). A satellite map view of this study’s monitoring setup is shown in Figure 2. Each of the eight bus stops used by participants are located in a large parking lot, alongside a major highway, or a combination of the two. 

### 2.4. Data Management

To measure and download PM_2.5_ concentrations, the AirBeam monitors must be used in conjunction with the free AirCasting mobile application. Participants downloaded the AirCasting application on their personal mobile phones from the Google Play Store or Apple App Store prior to beginning their study period. During their initial training, participants were instructed on the proper procedure for pairing the AirBeam monitor with the AirCasting application and recording air quality measurements. For ease of use, the application guides the user through the process of pairing the AirBeam monitor and recording a measurement session with on-screen prompts. Participants shared their AirCasting measurements with the research team via email daily sent directly from the application. The AirCasting application transmits session data records in a CSV file format. 

### 2.5. Data Analysis

AirCasting measurement session records were received from participants in a comma-separated values (CSV) file format. The manufacturer identification code of each respective AirBeam monitor was included in the session file, enabling the research team to determine which participant sent each file. Once identified, session files were converted to XSLX format to avoid data loss and stored in designated digital storage folders for each participant. PM_2.5_ concentration data from the EPA FEM and the ADR-1500 were included in these spreadsheets and in creating figures where applicable. Box plots of the bus stops with the EPA and ADR-1500 data were plotted with outliers. The primary objective of this study was to determine the personal PM_2.5_ exposure of various student participants during the time they typically spend waiting at an ECU Transit bus stop. To determine each participant’s personal PM_2.5_ exposure over the course of their study period, data were first separated into columns based on the participant from which they originated. Each participant was assigned a participant identification tag to protect anonymity, and a box plot was produced to detail each participant’s personal exposure.

Wind speed effects were also assessed to better understand the effects of high values on the low-cost sensor measurements. However, information was not collected about whether the student was waiting in an open/ unenclosed area or inside the bus stop shed (which could minimize wind effects). The average wind speeds for the county were only reported from https://www.wunderground.com (accessed on 1 July 2024). The humidity effects were not considered since this study was performed during the low humidity (<50%) season in Greenville, North Carolina, and humidity effects are usually observed above this value [35]. 

### 2.6. Analysis of Excessive Exposure

To better understand the excessive PM_2.5_ exposures the students had beyond the environment, hourly PM_2.5_ concentrations were obtained from the AirBeam monitors, the EPA FEM site, and the ADR-1500 monitors between 00:00 and 23:00 each day by averaging the corresponding minutely concentrations. Instances when the hourly PM_2.5_ concentrations from the AirBeam monitors were at least 5 units and 25% above the hourly PM_2.5_ concentrations from the EPS FEM site or the ADR-1500 monitors were counted and compared among bus stops and students. A limit of at least 5 units and 50% above the EPS FEM site or the ADR-1500 concentrations was also considered, but results were very similar to those of the 25% limit and not reported. Finally, the hourly PM_2.5_ concentrations from the AirBeam monitors were statistically compared among the bus stops and students using analysis of variance (ANOVA) after a logarithmic transformation to the data (to correct the skewness in data distributions).

## 3. Results

### 3.1. Bus Stop PM_2.5_ Concentrations

PM_2.5_ concentration measurements from each bus stop location throughout the entire study are depicted in the box plot in Figure 3. The minimum and maximum concentrations from each bus stop and the EPA FEM site and ADR-1500 monitors in February, March, and April are shown in Table 2. 

The data are shown alongside the PM_2.5_ concentration measurements reported by the Pitt County EPA FEM site and the ADR-1500. The figure includes outlier points that represent the concentrations at each bus stop on various days during each month, as each measurement received from participants represents a separate instance of waiting at a bus stop. Participant AirBeam data from bus stops were compared to PM_2.5_ concentration data from the EPA FEM site and ADR-1500. The average wind speeds for Greenville, NC, USA, between February 15 and April 17 ranged between 0 and 7.6 m/s. However, 70% of the data were below 3 m/s, and only 6 days were above 4 m/s. 

Compared to the EPA site, bus stops 1, 2, and 3 exceeded the mean PM_2.5_ concentration value reported by the Pitt County EPA FEM site in February. Bus stops 1, 3, 5, and 6 exceeded the maximum concentration value reported by the EPA FEM site in February. In March, only bus stop 6 exceeded the mean PM_2.5_ concentration value reported by the EPA FEM site. However, bus stops 1, 2, 4, 5, 6, and 7 each reported maximum concentration values that exceeded the maximum value reported by the EPA FEM site in March. In April, bus stops 2, 3, 4, and 5 reported a mean PM_2.5_ concentration value that exceeded the mean value reported by the EPA FEM site. Bus stops 2, 3, 4, 5, and 6 reported a higher maximum concentration value in April than was reported by the EPA FEM site.

Data were not collected from the ADR-1500 in April due to the limited availability of the research team responsible for the use and maintenance of the monitor, but the data span 15 February 2023 to 31 March 2023. The figure shows that bus stops 1, 2, and 3 each exceeded the mean PM_2.5_ concentration values reported by the ADR-1500 in February. Bus stops 1, 3, 5, and 6 each exceeded the maximum concentration value reported by the ADR-1500 in February. In March, none of the bus stops exceeded the maximum PM_2.5_ concentration values reported by the ADR-1500. However, bus stops 1, 2, 4, and 6 exceeded the maximum concentration value reported by the ADR-1500 in March. 

### 3.2. Examination of Participants’ Personal PM_2.5_ Exposure

The personal exposure for each participant compared to EPA and ADR-1500 monitors is shown in Figure 4. Outlier points were included to represent potential differences in PM_2.5_ concentration based on the bus stop locations at which each participant recorded measurements. Additionally, the outlier points represent the possibility of high PM_2.5_ concentrations in certain locations. The PM_2.5_ concentrations of participants 3 and 5 exceeded EPA and ADR-1500 concentrations. The PM_2.5_ concentrations for the other participants were similar to the reference instruments. 

### 3.3. Excessive PM_2.5_ Exposure

Table 3 summarizes the hourly PM_2.5_ concentrations. The average PM_2.5_ concentrations from the EPA FEM site and the ADR-1500 monitors were 7.85 (5.42) and 11.0 (6.02) µg/m^3^, respectively. All measurements from the AirBeam monitors had an average hourly PM_2.5_ concentration of 5.10 (5.38) µg/m^3^, which was lower than those from the EPS FEM site and the ADR-1500 monitors. The average hourly PM_2.5_ concentrations from the AirBeam monitors varied from 1.41 to 6.41 µg/m^3^ among the bus stops and from 3.83 to 6.33 µg/m^3^ among the students. Among the total 113 hourly PM_2.5_ concentrations from the AirBeam monitors, six (5.31%) were at least 5 units and 25% above the EPA FEM site concentrations, while five (4.42%) were at least 5 units and 25% above the ADR-1500 concentrations. These instances were quite evenly distributed across bus stops and students with each bus stop, and each student had 0–2 hourly concentrations above the limits. ANOVA analyses also revealed no statistically significant differences (*p* ≥ 0.65) in mean hourly PM_2.5_ concentrations among the bus stops and the students as measured by the AirBeam monitors.

## 4. Discussion

The PM_2.5_ mass concentrations at the bus stops exceeded the reference instruments on multiple occasions. Compared to the EPA FEM site, three of the seven bus stops reported higher mean PM_2.5_ concentrations in February, one bus stop reported higher mean PM_2.5_ concentrations in March, and four bus stops reported higher mean PM_2.5_ concentrations in April. Regarding maximum concentration values, four of the seven bus stops in February, six in March, and five in April reported a higher peak PM_2.5_ concentration value than was reported by the EPA FRM monitoring site. In addition, despite the ADR-1500 location at a busy road intersection, three of the bus stops in February reported higher average PM_2.5_ concentrations, and half of the bus stops in this study saw higher peak PM_2.5_ concentrations for two consecutive months compared to the ADR-1500. These results indicate support for the conclusion that students who use ECU Transit may face a mild-to-moderate risk of exposure to higher-than-average PM_2.5_ concentrations and a moderate-to-severe risk of exposure to peak concentrations higher than those reported from the Pitt County EPA FRM site and the ADR-1500 monitor.

The wind speeds for the duration of this study were mostly below 3 m/s. Ouimette et al. [36] showed that wind speeds lower than 3 m/s have little effect on the low-cost sensor PMS5003 measurements when located inside a weatherproof monitor. The PMS5003 is from the same manufacturer of the sensor used in the AirBeam monitor. Therefore, this can be interpreted in a similar manner for the AirBeam monitor, where only 30% of the days might have been affected by higher wind speeds, but only 6 days could have been omitted from the bus study due to high wind speeds (>4 m/s). However, the study did not report if the student was inside the bus stop shed. Therefore, it was not possible to omit these days.

Throughout the entire study period, none of the eight bus stop locations reported a PM_2.5_ concentration value that exceeded the EPA’s maximum 24 h limit of 35 µg/m^3^. However, the results of this study identified the possibility that PM_2.5_ concentrations for personal exposure can reach levels close to that standard. The EPA also has in place a long-term annual exposure limit of 12 µg/m^3^ for primary emissions, and while the duration of this study does not allow for the provision of PM_2.5_ concentrations for a year, numerous individual participant measurements reported mean concentrations that exceeded 12 µg/m^3^. According to study results, bus stop 3 reported a mean PM_2.5_ concentration of 12 µg/m^3^ over the course of the study period. Although nearly all the bus stops examined in this study reported maximum PM_2.5_ concentration values higher than 12 µg/m^3^ during February, March, and April, none of the bus stops reported a mean concentration higher than 12 µg/m^3^. True comparison to the 12 µg/m^3^ annual standard, however, would require this research to be conducted over the course of 12 months to better understand the seasonal impacts on PM_2.5_. 

To understand the PM_2.5_ concentrations reported from the bus stops better, the area in which the bus stop is located should be considered. Therefore, four of the three bus stops are discussed. For example, bus stop 1 (BS1), the Main Campus Student Center (MCSC) stop, is situated in a roundabout directly off a four-lane road beside the MCSC. Aside from a dormitory on one side and an ECU police station on another, the area is relatively open. Additionally, only one ECU Transit bus at a time may stop at BS1. Mean PM_2.5_ concentrations at BS1 in February and March were 10 and 3 µg/m^3^, respectively. The maximum concentration reported from BS1 was 15 µg/m^3^ in February and 18 µg/m^3^ in March. In contrast, bus stop 2 is an ECU Transit “hub” at which multiple buses may stop at any given time. BS2 is located off the same four-lane road as BS1, although the area in which BS2 is situated is less open compared to BS1. In February, March, and April, BS2 reported mean PM_2.5_ concentrations of 8, 4, and 8 µg/m^3^, respectively. The maximum concentration reported from BS2 was 10 µg/m^3^ in February, 30 µg/m^3^ in March, and 12 µg/m^3^ in April. Although BS2′s reported mean concentration in February was lower than that of BS1 in the same month, BS2 reported more sustained mean concentrations. This could be due to the frequent presence of multiple ECU Transit buses at a time, the relatively closed area in which BS2 is situated, or BS2 being directly to the side of a major four-lane road. BS3 is situated in the center of ECU’s largest student residential neighborhood. BS3 hosts six student dormitories, one of ECU’s two dining halls, a café, and an ECU Transit bus stop. BS3 reported mean PM_2.5_ concentrations of 12, 3, and 10 µg/m^3^ in February, March, and April, respectively. The maximum concentration reported from BS3 was 16 µg/m^3^ in February, 9 µg/m^3^ in March, and 17 µg/m^3^ in April. College Hill is densely populated with buildings and trees, and trees are known to remove a portion of ambient particulate matter through accumulation on plant surfaces [37]; however, the presence of several multi-story buildings may contribute to an obstruction of airflow in the area. Additional contributing factors to PM_2.5_ concentration in the area of BS3 may include the presence of several large parking lots and the frequent use of the roads near BS3 by drivers to travel quickly between two four-lane main streets. Given the dense student population at BS3, PM_2.5_ concentration may be especially affected by vehicle emissions during dormitory move-in and move-out periods, both of which occur multiple times per semester. Finally, bus stop 4 (BS4) is situated near a densely populated residential area for on-campus students. BS4 is located between multiple dormitories near a roundabout and a parking lot. Mean PM_2.5_ concentrations reported from BS4 were 2, 5, and 8 µg/m^3^ in February, March, and April, respectively. The maximum concentration reported from BS4 was 5 µg/m^3^ in February, 18 µg/m^3^ in March, and 17 µg/m^3^ in April. Potential contributing factors to PM_2.5_ concentrations in the area of BS4 could include an increase in traffic or frequent construction and road maintenance in the area. 

The statistical analyses suggested no significant differences in mean hourly PM_2.5_ exposures among the bus stop and the students. The analysis also suggested a relatively low (0–10.5%) chance of excessive PM_2.5_ exposures beyond the environment at the bus stops and a relatively low (0–13.3%) change in excessive PM_2.5_ exposures beyond the environment for each student. These results are not surprising as Greenville, NC, is a small city without much traffic and other PM_2.5_ sources. However, given that each student only recorded 11–25 h of data, these results should be interpreted carefully. More data may be needed to reach a more definite conclusion.

This study utilized the AirBeam2 and AirBeam3, the latter being the successor to the former. While the accuracy of both generations of the AirBeam personal aerosol monitor has been examined and proven by past research, Sousan, Regmi, and Park [23] found that the AirBeam2 consistently underestimated particulate matter concentrations in environmental settings. HabitatMap, manufacturer of the AirBeam monitors, states that the AirBeam3 is the most accurate model of the AirBeam monitor to date [25]. A previously referenced study performed by the South Coast Air Quality Management District found that the AirBeam3 was 78.6%-to-97% accurate for PM_2.5_ measurements in a laboratory evaluation, although it consistently underestimated PM_2.5_ concentrations compared to an EPA T640x FEM monitor [26,28]. However, the AirBeam3 showed a “strong to very strong” correlation to T640x and GRIMM (another FEM monitor) measurements in field evaluations [28].

### Limitations and Future Research

Due to time constraints and limited availability of monitoring equipment and funding, the research team was only able to recruit seven participants. While the small sample size did not necessarily impede or reduce data collection, additional participants would have potentially enabled the collection of data from a greater number of bus stops. The bus stops were selected based on the typical daily schedules of the seven participants, and additional participants would likely allow for the examination of other bus stops outside of the eight stops used in this study. In addition, due to the limited equipment availability and a lack of available funding, the study team utilized both generations (2 and 3) of the AirBeam monitor to allow for the largest sample size possible. Future implementations of this study may involve exclusive use of the AirBeam3 personal aerosol monitor. The data collection procedure from the current study does not allow for the accurate comparison of PM_2.5_ concentration measurements between the two generations of the AirBeam used. Inter-sensor performance comparisons and calculations of the limit of detection through the co-location of AirBeam2 and AirBeam3 monitors were not performed in this study. However, this study took advantage of the high correlation between AirBeam models and reference instruments previously determined by SCAQMD [26,28]. This study could be continued with an updated procedure that enables this comparison via co-location of an AirBeam2 and AirBeam3 monitor at a fixed bus stop location. In addition, future work could monitor the bus stops directly and reassess possible exposures at different locations and times during the day. Future studies should also take into account high wind speed effects by evaluating these effects on the performance of the low-cost sensor near a reference instrument. 

Future implementation of this study would involve a significantly larger number of participants, contingent on the research team’s ability to secure additional project funding. An interesting future direction for this study could see the comparison of the AirBeam monitor with several other low-cost monitors at bus stops in real environmental conditions. Assuming that these factors were not limiting, an interesting new direction of this study could be a year-long analysis of PM_2.5_ concentrations at ECU Transit bus stops. Collecting data for an entire calendar year would allow the study team to better analyze the effects of environmental conditions on PM_2.5_ concentration readings and identify seasonal, time-based trends in personal exposure. 

## 5. Conclusions

The results of this study did not indicate that exposure to PM_2.5_ concentrations higher than the EPA average for Pitt County was a continuously present risk. Concentration values reported from bus stop locations were often close to those reported by the EPA site on numerous occasions, and extreme elevations in concentration compared with EPA averages were relatively uncommon. In addition, this study indicated that mean PM_2.5_ concentrations shifted over time similarly to concentrations reported by the EPA FRM site, but instances of abnormally high PM_2.5_ concentration were reported on numerous occasions at several different bus stop locations. The possibility of external factors must be considered, as some sort of event may be occurring near a bus stop location that is not occurring in the vicinity of the EPA FRM site. It is known that personal monitoring can provide more individually relevant information than an ambient monitoring site several kilometers away from one’s location. Students who commute via the ECU Transit bus system may face a mild-to-moderate risk of exposure to PM_2.5_ concentrations that are higher than those reported from the EPA FRM monitoring site used to monitor county-level exposures. Further, these students may also face a moderate-to-severe risk of exposure to higher peak PM_2.5_ concentrations than those reported by the EPA FRM site. Regardless, the results of this study highlighted the importance of ECU students’ awareness of air quality and the potential impacts it may have on their health. The results of this study have the potential to serve as an outlet through which ECU students who use the ECU Transit system can obtain more personally relevant air quality information.

## Figures and Tables

**Figure 1 sensors-24-04520-f001:**
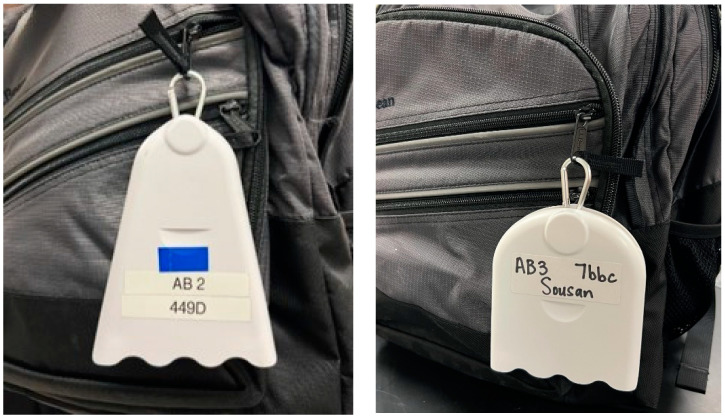
AirBeam2 (**left**) and AirBeam3 (**right**) attached to a student’s book bag.

**Figure 2 sensors-24-04520-f002:**
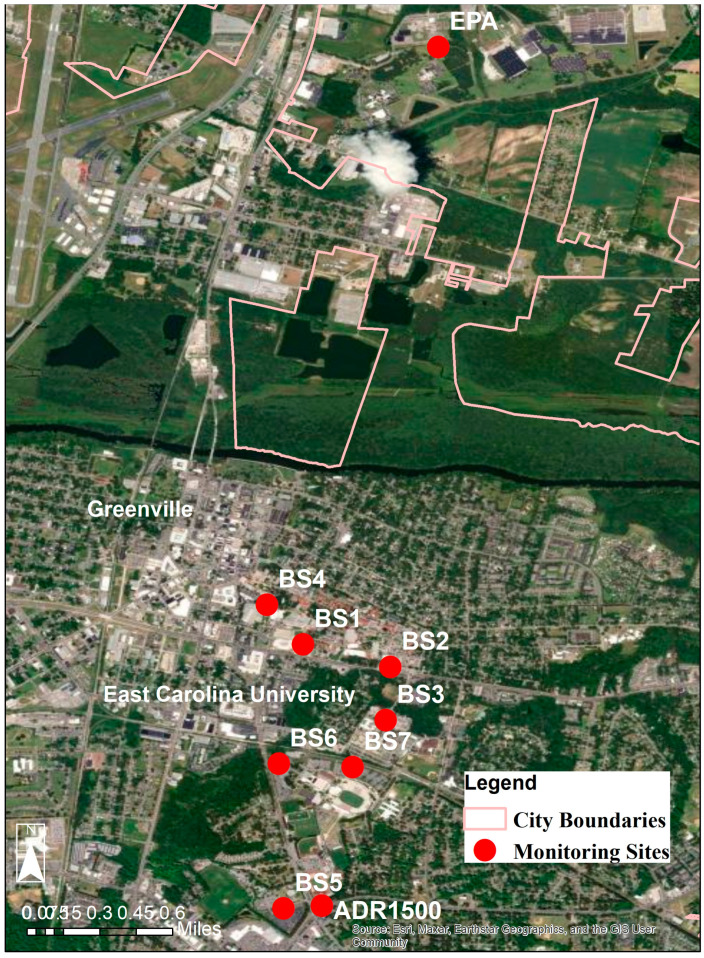
Map of this study’s monitoring locations of each of the eight ECU Transit bus stop locations used by participants and the reference instruments.

**Figure 3 sensors-24-04520-f003:**
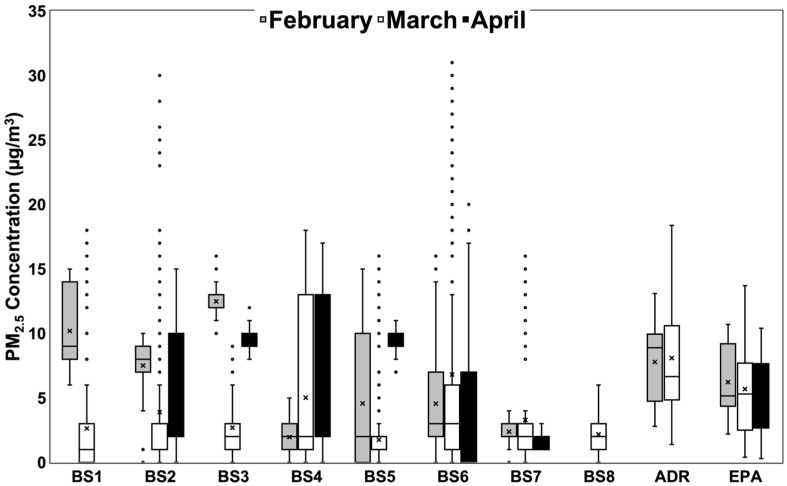
Box plot of PM_2.5_ concentration measurements from the eight bus stops included in this study, as well as the EPA FEM site and the ADR-1500, with outlier points included.

**Figure 4 sensors-24-04520-f004:**
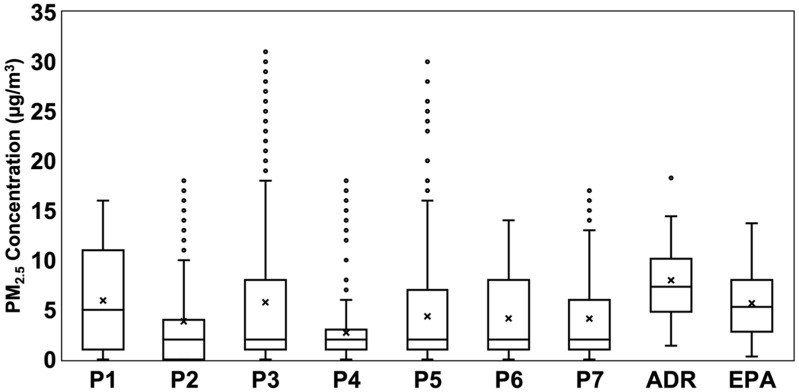
Personal PM_2.5_ exposures of each participant from all bus stops used, shown alongside concentration data from the ADR-1500 and Pitt County EPA FEM sites.

**Table 1 sensors-24-04520-t001:** The date range when the participant was collecting data.

ID	Period
P1	15 February 2023–16 March 2023
P2	17 February 2023–17 March 2023
P3	27 February 2023–27 March 2023
P4	20 March 2023–17 April 2023
P5	20 March 2023–17 April 2023
P6	20 March 2023–17 April 2023
P7	20 March 2023–17 April 2023

**Table 2 sensors-24-04520-t002:** Mean and maximum PM_2.5_ concentrations from each bus stop and the EPA FEM site and ADR-1500, in µg/m^3^, from February, March, and April. Bus stop concentrations that exceeded EPA and ADR-1500 concentration measurements within each respective parameter are highlighted.

Bus Stop ID	February Mean	February Max	March Mean	March Max	April Mean	April Max
1	10	15	3	18	-	-
2	8	10	4	30	7	15
3	12	16	3	9	10	12
4	2	5	5	18	8	17
5	5	15	2	16	9	11
6	5	16	7	31	4	20
7	2	4	3	16	2	3
8	-	-	2	6	-	-
EPA Site	6	11	6	14	5	10
ADR-1500	8	13	8	18	-	-

**Table 3 sensors-24-04520-t003:** Summary of hourly PM_2.5_ concentrations.

Student/ Bus Stop/Site	N	Mean (µg/m^3^)	SD (µg/m^3^)	n (%) ≥5 Units and 25% above EPA	n (%) ≥5 Units and 25% above ADR
ALL	113	5.10	5.38	6 (5.31)	5 (4.42)
BS1	12	5.30	5.40	1 (8.33)	0 (0)
BS2	24	5.78	6.47	2 (8.33)	2 (8.33)
BS3	11	5.31	3.86	0 (0)	0 (0)
BS4	19	4.69	5.25	1 (5.26)	2 (10.5)
BS5	18	3.90	4.52	1 (5.56)	0 (0)
BS6	19	6.41	6.36	1 (5.26)	1 (5.26)
BS7	8	4.02	4.25	0 (0)	0 (0)
BS8	2	1.41	1.84	0 (0)	0 (0)
P1	11	5.40	4.91	0 (0)	0 (0)
P2	25	4.53	5.09	1 (4.00)	2 (8.00)
P3	15	5.28	7.01	1 (6.67)	2 (13.3)
P4	11	3.83	4.82	0 (0)	0 (0)
P5	24	6.33	6.24	2 (8.33)	1 (4.17)
P6	13	5.48	4.23	1 (7.69)	0 (0)
P7	14	4.26	4.60	1 (7.14)	0 (0)
EPA	113	7.85	5.42		
ADR	113	11.0	6.02		

## Data Availability

The data will be available upon request.

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
