# Peer review of "Assessment of PM2.5 Concentration at University Transit Bus Stops Using Low-Cost Aerosol Monitors by Student Commuters"

_sensors, 2024, doi:10.3390/s24144520_

Round 1

Reviewer 1 Report

Comments and Suggestions for Authors

The manuscript describes a descriptive study of students using mobile PM2.5 measuring devices. The study is relevant in the context of air quality. Relatively inexpensive particle detectors were used during the research.

The main remark - I missed the low cost detector calibration procedure. Are they adjusted in any way at all for their PM measurements? Also , low cost and ADR-1500 meters were not compared at the same point. I mean that probably is possible leave the devices e.g. two days next to each other and later make sure that the measurements coincide?

A minor note is to use SI units in the text.

Line 303. mistake: pDR-1500

Author Response

The main remark - I missed the low cost detector calibration procedure. Are they adjusted in any way at all for their PM measurements? Also , low cost and ADR-1500 meters were not compared at the same point. I mean that probably is possible leave the devices e.g. two days next to each other and later make sure that the measurements coincide?

  • Response to the reviewer: We agree with the reviewer that co-locating the ADR-1500 with the low-cost monitors would have provided informative statistical analysis. However, this was not performed for this study due to feasibility issues. Moreover, the ADR is currently not deployed in Greenville and was removed from the site. Therefore, due to the unavailability of funding, we are unable to perform any further experimental work at this time. Future work will attempt to secure more funding by using this work as pilot data and acquire funds to increase the sample size and perform side-by-side comparisons with high-cost and reference instruments. In addition, the following text was added to line 142 to further address the reviewer's comments and concerns:

Primarily, the AirBeam monitors have been calibrated by the manufacturer and extensively evaluated in outdoor environmental settings and have shown a high correlation (r2 ≥ 0.9) compared to reference instruments [25, 27]. Therefore, calibration was not performed for the sensors in the study. Moreover, performing such calibration that entails deploying a high-cost instrument with low-cost sensors for personal exposure was impractical for the study due to the cost of the instruments and the inability to train students to operate these high-cost monitors, defeating the purpose of performing citizen science research. The purpose of the study is to use these manufacturer-calibrated and well-evaluated low-cost sensors to better understand student exposure at the bus stop and show the differences by comparing these values with reference and high-cost instruments deployed in the same city and county while considering their limitation based on previous work. This is important because EPA sites represent county-level data, and this work shows that averaging data for the county presents biases.”

In addition, the following sentence was incorrect and was updated: from “However, this study utilized regression models derived from to ensure data accuracy, the study could be continued with an updated procedure that enables this comparison via co-location of an AirBeam2 and AirBeam3 monitor at a fixed bus stop location.” To “However, this study took advantage of the high correlation between AirBeam models and reference instruments previously by SCAQMD [25, 27]. The study could be continued with an updated procedure that enables this comparison via co-location of an AirBeam2 and AirBeam3 monitor at a fixed bus stop location.

A minor note is to use SI units in the text.

  • Response to the reviewer: The following changes were made
    • The sentence on line 125 was changed from “within 300 feet of the site” To “within 483 km of the site”
    • The sentence on line 453 was changed “site several miles away” To “site several kilometers away”

Line 303. mistake: pDR-1500

  • Response to the reviewer: The text was changed on line 303 from “pDR-1500 concentrations” To “ADR-1500 concentrations”

Other Changes:

Citations 11, 15, 16, 17, and 18 were replaced with the following citations, respectively. Lowering the number of self-citations by 50%.

  • Mehadi, A.; Moosmüller, H.; Campbell, D. E.; Ham, W.; Schweizer, D.; Tarnay, L.; Hunter, J., Laboratory and field evaluation of real-time and near real-time PM2.5 smoke monitors. Journal of the Air & Waste Management Association 2020, 70, (2), 158-179.
  • Tsameret, S.; Furuta, D.; Saha, P.; Kwak, N.; Hauryliuk, A.; Li, X.; Presto, A. A.; Li, J., Low-Cost Indoor Sensor Deployment for Predicting PM2.5 Exposure. ACS ES&T Air 2024, 533-545.
  • Li, J., Mattewal, S.K., Patel, S. and Biswas, P. (2020). Evaluation of Nine Low-cost-sensor-based Particulate Matter Monitors. Aerosol Air Qual. Res. 20: 254-270. https://doi.org/10.4209/aaqr.2018.12.0485

  • Wang, Y.; Li, J.; Jing, H.; Zhang, Q.; Jiang, J.; Biswas, P., Laboratory evaluation and calibration of three low-cost particle sensors for particulate matter measurement. Aerosol Sci. Technol. 2015, 49, (11), 1063-1077.

  • Kelly, K. E.; Whitaker, J.; Petty, A.; Widmer, C.; Dybwad, A.; Sleeth, D.; Martin, R.; Butterfield, A., Ambient and laboratory evaluation of a low-cost particulate matter sensor. Environmental Pollution 2017, 221, (Supplement C), 491-500.

Reviewer 2 Report

Comments and Suggestions for Authors

This study presents a random sampling study of bus stops at East Carolina University (ECU) in Greenville, North Carolina using low-cost sensors. Bus stop PM2.5 mass concentration anomalies were monitored based on the trips of seven volunteers, and student exposure to particulate matter emitted from traffic sources is discussed. The article has a completed structure and details of the study. However, I have a few minor suggestions:

(1)      An important issue with low-cost sensors is their low accuracy; the PM2.5 mass change values obtained from the measurements in this paper are around 1 μg (25%), and I think that the authors should have first confirmed that low-cost sensors can have sufficient monitoring capability for this concentration.

(2)      The study gives bus stop observations as waiting times for only 7 volunteers' journeys, corresponding to the EPA and ADR also obtained for comparison of response time results? This affects the difference in results due to sampling frequency.

(3)      The study did not discuss the meteorological conditions during the observation period, which can also have an impact on bus stop observations, e.g., at higher wind speeds, the PM2.5 mass concentrations at stops between buildings can be significantly different. Therefore, it is recommended that meteorological observations be carried out and the mechanism of their influence be discussed.

(4)      Despite the extensive discussion of the effects of human exposure in the study, I still believe that the data collected from this observation is insufficient and therefore, in order to obtain better results, the trial should indeed be expanded and the data analysed in further depth as stated by the authors. I would advise authors not to use "exposed" in the title to make it more rigorous.

Comments on the Quality of English Language

Minor editing of English language required

Author Response

(1)      An important issue with low-cost sensors is their low accuracy; the PM2.5 mass change values obtained from the measurements in this paper are around 1 μg (25%), and I think that the authors should have first confirmed that low-cost sensors can have sufficient monitoring capability for this concentration.

  • Response: Due to the unavailability of funding, we are unable to perform any further experimental work at this time. Future work will attempt to secure more funding by using this work as pilot data and acquire funds to increase the sample size and perform side-by-side comparisons with high-cost and reference instruments.
  • We agree the LOD values were not calculated. That is why we added the following sentence on line 417: “Inter-sensor performance comparisons and calculations of the limit of detection through the co-location of AirBeam2 and AirBeam3 monitors were not performed in this study.”
  • This project was based on limited funding and is the first of its kind in terms of sampling exposure for students at the bus stop during their daily commute. We understand the reviewer’s and we have added these concerns as limitations to our study, and we have added additional limitations based on your feedback. Despite these limitations, we believe the data in this work is valuable for publication and will help future work improve on the methods given the appropriate funding. Since funding opportunities are always difficult to acquire, other groups might perform future research using our work and limitations as guidelines. Thank you again for your understanding.

(2)      The study gives bus stop observations as waiting times for only 7 volunteers' journeys, corresponding to the EPA and ADR also obtained for comparison of response time results? This affects the difference in results due to sampling frequency.

  • Response: The study does take into account time pairing in sections 2.6 and 3.3, where the hourly PM2.5 concentrations are time-paired for the three different devices, and then the statistical analysis is performed.

(3)      The study did not discuss the meteorological conditions during the observation period, which can also have an impact on bus stop observations, e.g., at higher wind speeds, the PM2.5 mass concentrations at stops between buildings can be significantly different. Therefore, it is recommended that meteorological observations be carried out and the mechanism of their influence be discussed.

  • Response: The following sentences were added to address the comment:
    • Line 261: “Wind speed effects were also assessed to better understand the effects of high values on the low-cost sensor measurements. However, information was not collected about whether the student was waiting in an open/ unenclosed area or inside the bus stop shed (which could minimize wind effects). The average wind speeds for the county were only reported from https://www.wunderground.com. The humidity effects were not considered since the study was performed during the low humidity (<50%) season in Greenville, North Carolina, and humidity effects are usually observed above this value [35].
    • The following reference was added: [35] Tryner, J.; Mehaffy, J.; Miller-Lionberg, D.; Volckens, J., Effects of aerosol type and simulated aging on performance of low-cost PM sensors. J Aerosol Sci 2020, 150, 105654.
    • Line 293: “The average wind speeds for Greenville, NC, between February 15 and April 17 ranged between 0 and 7.6 m/s. However, 70% of the data was below 3 m/s, and only 6 days were above 4 m/s.”
    • Line 364: “The wind speeds for the duration of the study were mostly below 3 m/s. Ouimette, et al. [36] showed that wind speeds lower than 3 m/s have little effect on the low-cost sensor PMS5003 measurements when located inside a weatherproof monitor. The PMS5003 is from the same manufacturer of the sensor used in the AirBeam monitor. Therefore, this can be interpreted in a similar manner for the AirBeam monitor, where only 30% of the days might have been affected by higher wind speeds, but only 6 days could have been omitted from the study due to high wind speeds (> 4 m/s). However, the study did not report if the student was inside the bus stop shed. Therefore, it was not possible to omit these days.”
    • The following reference was added: [36] Ouimette, J.; Arnott, W. P.; Laven, P.; Whitwell, R.; Radhakrishnan, N.; Dhaniyala, S.; Sandink, M.; Tryner, J.; Volckens, J., Fundamentals of low-cost aerosol sensor design and operation. Aerosol Science and Technology 2024, 58, (1), 1-15.
    • Line 466: “Future studies should also take into account high wind speed effects by evaluating these effects on the performance of the low-cost sensor near a reference instrument.”

s

(4)      Despite the extensive discussion of the effects of human exposure in the study, I still believe that the data collected from this observation is insufficient and therefore, in order to obtain better results, the trial should indeed be expanded and the data analysed in further depth as stated by the authors. I would advise authors not to use "exposed" in the title to make it more rigorous.

  • Response: Due to the unavailability of funding, we are unable to perform any further experimental work at this time. Future work will attempt to secure more funding by using this work as pilot data and acquire funds to increase the sample size and perform side-by-side comparisons with high-cost and reference instruments. The small sample size was identified as a limitation of the study. This project was based on limited funding and is the first of its kind in terms of sampling exposure for students at the bus stop during their daily commute. We understand the reviewer’s concerns about the small number of participants recruited based on the project funding, among other concerns. We have added these concerns as limitations to our study, and we have added additional limitations based on your feedback. Despite these limitations, we believe the data in this work is valuable for publication and will help future work improve on the methods given the appropriate funding. Since funding opportunities are always difficult to acquire, other groups might perform future research using our work and limitations as guidelines. Thank you again for your understanding.
  • The title was changed to “Assessment of PM2.5 Concentration at University Transit Bus Stops Using Low-Cost Aerosol Monitors by Student Commuters

Other Changes:

Citations 11, 15, 16, 17, and 18 were replaced with the following citations, respectively. Lowering the number of self-citations by 50%.

  • Mehadi, A.; Moosmüller, H.; Campbell, D. E.; Ham, W.; Schweizer, D.; Tarnay, L.; Hunter, J., Laboratory and field evaluation of real-time and near real-time PM2.5 smoke monitors. Journal of the Air & Waste Management Association 2020, 70, (2), 158-179.
  • Tsameret, S.; Furuta, D.; Saha, P.; Kwak, N.; Hauryliuk, A.; Li, X.; Presto, A. A.; Li, J., Low-Cost Indoor Sensor Deployment for Predicting PM2.5 Exposure. ACS ES&T Air 2024, 533-545.
  • Li, J., Mattewal, S.K., Patel, S. and Biswas, P. (2020). Evaluation of Nine Low-cost-sensor-based Particulate Matter Monitors. Aerosol Air Qual. Res. 20: 254-270. https://doi.org/10.4209/aaqr.2018.12.0485

  • Wang, Y.; Li, J.; Jing, H.; Zhang, Q.; Jiang, J.; Biswas, P., Laboratory evaluation and calibration of three low-cost particle sensors for particulate matter measurement. Aerosol Sci. Technol. 2015, 49, (11), 1063-1077.

  • Kelly, K. E.; Whitaker, J.; Petty, A.; Widmer, C.; Dybwad, A.; Sleeth, D.; Martin, R.; Butterfield, A., Ambient and laboratory evaluation of a low-cost particulate matter sensor. Environmental Pollution 2017, 221, (Supplement C), 491-500.